# Association between *APOL1* risk variants and the occurrence of sepsis in Black patients hospitalized with infections: a retrospective cohort study

Lan Jiang[1], Ge Liu[2], Annette Oeser[1], Andrea Ihegword[1], Alyson L Dickson[1,3], Laura L Daniel[4], Adriana M Hung[5,6], Nancy J Cox[7], Cecilia P Chung[4], Wei-Qi Wei[2], C Michael Stein[1,8], Qiping Feng[1,7]*

[1]Division of Clinical Pharmacology, Department of Medicine, Vanderbilt University Medical Center, Nashville, United States; [2]Department of Biomedical Informatics, Vanderbilt University Medical Center, Nashville, United States; [3]Division of Rheumatology and Immunology, Department of Medicine, Vanderbilt University Medical Center, Nashville, United States; [4]Division of Rheumatology, Department of Medicine, University of Miami, Miami, United States; [5]Tennessee Valley Healthcare System, Nashville Campus, Nashville, United States; [6]Division of Nephrology & Hypertension, Vanderbilt University Medical Center, Nashville, United States; [7]Vanderbilt Genetics Institute, Department of Medicine, Vanderbilt University Medical Center, Nashville, United States; [8]Department of Pharmacology, Vanderbilt University, Nashville, United States

*For correspondence:
qiping.feng@vumc.org

Competing interest: The authors declare that no competing interests exist.

## Abstract

**Background:** Two risk variants in the apolipoprotein L1 gene (*APOL1*) have been associated with increased susceptibility to sepsis in Black patients. However, it remains unclear whether *APOL1* high-risk genotypes are associated with occurrence of either sepsis or sepsis-related phenotypes in patients hospitalized with infections, independent of their association with pre-existing severe renal disease.

**Methods:** A retrospective cohort study of 2242 Black patients hospitalized with infections. We assessed whether carriage of *APOL1* high-risk genotypes was associated with the risk of sepsis and sepsis-related phenotypes in patients hospitalized with infections. The primary outcome was sepsis; secondary outcomes were short-term mortality, and organ failure related to sepsis.

**Results:** Of 2242 Black patients hospitalized with infections, 565 developed sepsis. Patients with high-risk *APOL1* genotypes had a significantly increased risk of sepsis (odds ratio [OR]=1.29 [95% CI, 1.00–1.67; p=0.047]); however, this association was not significant after adjustment for pre-existing severe renal disease (OR = 1.14 [95% CI, 0.88–1.48; p=0.33]), nor after exclusion of those patients with pre-existing severe renal disease (OR = 0.99 [95% CI, 0.70–1.39; p=0.95]). *APOL1* high-risk genotypes were significantly associated with the renal dysfunction component of the Sepsis-3 criteria (OR = 1.64 [95% CI, 1.21–2.22; p=0.001]), but not with other sepsis-related organ dysfunction or short-term mortality. The association between high-risk *APOL1* genotypes and sepsis-related renal dysfunction was markedly attenuated by adjusting for pre-existing severe renal disease (OR = 1.36 [95% CI, 1.00–1.86; p=0.05]) and was nullified after exclusion of patients with pre-existing severe renal disease (OR = 1.16 [95% CI, 0.74–1.81; p=0.52]).

**Conclusions:** *APOL1* high-risk genotypes were associated with an increased risk of sepsis; however, this increased risk was attributable predominantly to pre-existing severe renal disease.

**Funding:** This study was supported by R01GM120523 (QF), R01HL163854 (QF), R35GM131770 (CMS), HL133786 (WQW), and Vanderbilt Faculty Research Scholar Fund (QF). The dataset(s) used for the analyses described were obtained from Vanderbilt University Medical Center's BioVU which is supported by institutional funding, the 1S10RR025141-01 instrumentation award, and by the CTSA grant UL1TR0004from NCATS/NIH. Additional funding provided by the NIH through grants P50GM115305 and U19HL065962. The authors wish to acknowledge the expert technical support of the VANTAGE and VANGARD core facilities, supported in part by the Vanderbilt-Ingram Cancer Center (P30 CA068485) and Vanderbilt Vision Center (P30 EY08126). The funders had no role in design and conduct of the study; collection, management, analysis, and interpretation of the data; preparation, review, or approval of the manuscript; and decision to submit the manuscript for publication.

## eLife assessment

In this **valuable** study, patients homozygous for both minor frequency alleles of the APOL1 gene are shown to be at significant risk for progression into sepsis after infection. The study has enrolled a significant number of subjects and provides **solid** results. The study addresses to infectious diseases and critical care experts and one major weakness is the lack of inclusion of non-Black patients.

## Introduction

Sepsis is a common cause of morbidity and mortality in the United States, accounting for one in every two to three deaths that occur in hospitals (*Martin et al., 2003*). The risk of sepsis and associated mortality are approximately 60% and 80% higher, respectively, for Black patients compared to White patients (*Martin et al., 2003*; *Prest et al., 2021*). The higher incidence of sepsis in Black individuals persists after adjustment for comorbidities and socioeconomic status, encompassing both a greater risk for developing infection, and once infected, a greater risk of organ dysfunction (*Mayr et al., 2010*). Recent work in the Million Veteran Program (MVP) found that variants in the apolipoprotein L1 gene (*APOL1*), common in people of African ancestry, are associated with sepsis incidence and severity (*Wu et al., 2021*).

Two genetic variants in *APOL1*—termed G1 (rs73885319/rs60910145) and G2 (rs71785313)—are found almost exclusively in individuals of African ancestry and confer resistance to *Trypanosoma brucei* infection (*Genovese et al., 2010*; *O'Toole et al., 2017*; *Tzur et al., 2010*; *Lipkowitz et al., 2013*). However, individuals carrying two such alleles (i.e., G1/G1, G1/G2, or G2/G2) have a marked increase in the risk of chronic renal disease; for example, among African-Americans, individuals carrying two *APOL1* risk alleles are 7.3 times more likely to develop hypertension-associated end-stage renal disease (ESRD) compared to those without two risk alleles (*Genovese et al., 2010*). Correspondingly, carriage of two risk alleles is associated with an increased prevalence of numerous renal-related disorders, including hypertension, focal segmental glomerulosclerosis, and HIV-associated nephropathy (*Foster et al., 2013*; *Freedman et al., 2018*; *Kopp et al., 2011*). While these high-risk alleles typically follow a pattern of recessive expression (i.e., without increased risk for individuals carrying a single allele), because their carriage rates are so high among African-Americans (10–15% of African-Americans carry two *APOL1* risk alleles), a substantial portion of this population faces increased *APOL1*-related risk (*Genovese et al., 2010*).

The specific mechanisms whereby the *APOL1* risk variants increase the risk for renal disease are not fully understood, but beyond their role in innate immunity and resistance to trypanosomiasis, the *APOL1* risk variants are considered gain-of-injury variants (*Kruzel-Davila et al., 2016*), intensifying autophagy, cell death, endothelial cell inflammation and dysfunction, and immune pathway activation (*Wu et al., 2021*; *Friedman and Pollak, 2020*; *Daneshpajouhnejad et al., 2022*). However, many patients with two *APOL1* risk variants never develop renal disease, leading researchers to propose a two-hit model in which genetic susceptibility combined with an inflammatory trigger leads to disease (*Kruzel-Davila et al., 2016*). Indeed, *APOL1* expression is induced by inflammatory cytokines such as tumor necrosis factor alpha and gamma-interferon (*Limou et al., 2015*; *Nichols et al., 2015*); thus increased transcription of the gain-of-function variant *APOL1* is most likely to occur in the setting of severe infection.

**eLife digest** When the body is fighting off an infection, the processes it uses to protect itself can sometimes overreact. This results in a condition known as sepsis which can cause life-threatening damage to multiple organs. In the United States, Black patients are 60-80% more likely to develop sepsis compared to individuals who identify as White; differences remain even after accounting for socio-economic status and presence of other illnesses.

Recent work has suggested that two variants of the *APOL1* gene which are almost exclusively found in people with African ancestry may be a contributing factor to this disparity. These 'high-risk' genetic variants have also been shown to increase the likelihood of kidney diseases. It is therefore possible that the elevated chance of sepsis is not directly linked to these variations of *APOL1*, but rather is the result of patients already having reduced kidney function.

To understand the relationship between *APOL1* and sepsis, Jiang et al. analyzed data from patients admitted to Vanderbilt University Medical Centre in the United States between 2000 and 2020. This included 2,242 patients who identified as Black and had been hospitalized with an infection. The analyses showed that 16% of these individuals were carriers of the *APOL1* high-risk variants.

The high-risk patients were more likely to experience sepsis and demonstrate kidney damage. But other organs commonly damaged by sepsis were not affected more in these individuals compared to the other 84% of patients who did not have these variants. Furthermore, when individuals with pre-existing kidney diseases were removed from this high-risk group, the increased likelihood of sepsis was no longer prominent.

These findings suggest that the *APOL1* variants do not directly increase the risk of sepsis, and this association is primarily due to patients with these genetic variations being more susceptible to kidney diseases. There are new drugs under development targeting the *APOL1* variants. While these may provide protection against kidney diseases, they are unlikely to be successful at preventing or treating sepsis once a patient has been hospitalized with an infection.

Given the substantive risks and costs associated with sepsis, as well as the reported association between sepsis and the presence of two *APOL1* risk alleles (*Wu et al., 2021*), it is critical to better understand whether *APOL1* has a direct association with sepsis. More specifically, since *APOL1* is associated with renal disease (*Foster et al., 2013*; *Bajaj et al., 2020*; *Yusuf et al., 2021*; *Chen et al., 2021*) and existing renal disease is associated with worse sepsis outcomes (*Doi et al., 2008*) (including increased kidney injury), the relationship between *APOL1* and sepsis external to existing renal disease remains unclear. Further, it is important to assess whether other organ system dysfunctions typical of sepsis (i.e., hepatic, respiratory, circulatory, and hematologic dysfunction) are associated with *APOL1* to appropriately define the causal pathway between *APOL1* and sepsis. This determination of association could help guide the viability of implementing treatment options for sepsis using therapies that target APOL1. In particular, several inhibitors of APOL1 are in various stages of development—if *APOL1* high-risk genotypes are directly associated with the pathogenesis of sepsis external to pre-existing severe renal disease or other organ system dysfunction, such drugs could be offered to high-risk patients to prevent or treat the occurrence of sepsis in scenarios of acute infection (*Friedman et al., 2022*).

The objective of this study was to better understand the relationship between *APOL1* and sepsis, focusing on two critical points. First, whether *APOL1* high-risk genotypes are associated with the risk of sepsis for patients hospitalized with an infection, particularly independent of the association between the genotypes and severe renal disease. Second, among patients carrying *APOL1* high-risk genotypes, whether the risk of organ dysfunction that defines the presence of sepsis is limited to renal dysfunction *or* if it affects other typical organ system dysfunction components of sepsis.

## Methods
### Study population and design
This study used data from the Vanderbilt University Medical Center (VUMC) Synthetic Derivative, which contains a de-identified version of the electronic medical records (EHRs) for VUMC patients

(~3.6 million individual records as of October 2022). These de-identified EHRs are linked to a biobank (BioVU), which has genome-wide genotyping results for ~120,000 patients. From these genotyped patients, we constructed a cohort of Black patients admitted to the hospital with an infection to assess the association between carriage of *APOL1* high-risk alleles and the occurrence of sepsis before and after consideration of pre-existing severe renal disease. Additionally, to replicate findings from a previously published MVP sepsis study (*Wu et al., 2021*), we performed a restricted phenome-wide association analysis (PheWAS) in all Black patients with existing genotypes; we determined the association between carriage of *APOL1* high-risk genotypes and phenotypes previously reported to be associated with sepsis, further expanding the analysis to account for pre-existing severe renal disease based upon the primary analysis of this study (*Wu et al., 2021*). This study was reviewed by the VUMC Institutional Review Board; given the study's retrospective design and use of de-identified data only, informed consent was waived.

## Inclusion/exclusion criteria

The primary cohort included individuals with EHR-reported Black race who were admitted to the hospital with an infection between January 2000 and August 2020 and were ≥18 years of age on the day of admission (*Liu et al., 2023*). We selected patients with EHR-reported Black race only (i.e., as reported by the patient or, secondarily, by a provider) because *APOL1* high-risk genotypes are virtually exclusive to populations of recent African ancestries (*Kopp et al., 2011*; *Daneshpajouhnejad et al., 2022*; *Limou et al., 2014*; *Zhang et al., 2016*) the majority of whom are identified as having Black race in our dataset (i.e., EHR-reported race is highly consistent with genetic ancestry in BioVU) (*Dumitrescu et al., 2010*). This restriction helps control for factors associated with socially determined race which may affect individuals' sepsis outcomes external to genetic ancestry (e.g., an individual with predominantly African genetic ancestry who identifies as having White race). To account for genetic ancestral diversity within the cohort, all analyses were adjusted for principal components (PCs) for ancestry (see below). The day of hospital admission was designated day 0. International Classification of Disease, ninth revision, Clinical Modification (ICD-9-CM); tenth revision (ICD-10-CM); Current Procedural Terminology (CPT) codes; medications; labs; and clinical notes were used for cohort construction and covariates. Infection was defined as having a billing code indicating an infection and receiving an antibiotic within 1 day of hospital admission (i.e., on days –1, 0, or +1) (*Liu et al., 2023*; *Feng et al., 2019a*; *Feng et al., 2019b*). We used ICD-9-CM and ICD-10-CM codes for this definition of infection based on the criteria of *Angus et al., 2001*; *Donnelly et al., 2017*, excluding viral, mycobacterial, fungal, and spirochetal infections, as we have described in detail previously (*Liu et al., 2023*; *Feng et al., 2019a*; *Feng et al., 2019b*). Only the first hospitalization for infection was included if a patient had more than one qualifying episode. We excluded individuals admitted for cardiac surgery, cardiogenic shock, and organ transplantation, as well as those with no relevant laboratory values (i.e., creatinine, bilirubin, or platelets) on days –1, 0, or +1. We also excluded patients who had a positive test or ICD-10-CM code (U07.1) for coronavirus disease (COVID-19) on days –1, 0, or +1 (*Liu et al., 2023*; *Feng et al., 2019a*; *Feng et al., 2019b*).

## Outcomes

The primary outcome was the development of sepsis as indicated by fulfillment of the Sepsis-3 criteria (described below). Secondary outcomes were the individual organ dysfunction criteria in the Sepsis-3 definition (i.e., renal, hepatic, respiratory, circulatory, and hematologic dysfunction) as well as severe sepsis/septic shock and short-term mortality.

Sepsis was defined by the Sepsis-3 criteria of concurrent infection and organ dysfunction (*Liu et al., 2023*; *Donnelly et al., 2017*; *Rhee et al., 2017*) using the EHR definition that was developed in real-world hospital settings (*Rhee et al., 2017*; *Seymour et al., 2016*) and optimized, validated, and applied across EHR systems from 409 hospitals (*Rhee et al., 2017*). The algorithm uses billing codes and clinical criteria, with a specificity of 98.1% and a sensitivity of 69.7% (*Rhee et al., 2017*). We further adapted (i.e., included relevant ICD-10-CM codes) and then applied the EHR-based Sepsis-3 algorithm to the de-identified EHRs in our system, as described previously (*Feng et al., 2019a*; *Feng et al., 2019b*). Because the vast majority of community acquired sepsis cases (87%) are present on admission to hospital (*Rhee et al., 2017*), we studied sepsis occurring within 1 day of hospital admission (days –1, 0, and +1) to minimize the confounding effects of sepsis occurring secondary to

procedures or events in the hospital. We previously validated this updated algorithm in our EHRs (*Liu et al., 2023*; *Feng et al., 2019a*; *Feng et al., 2019b*).

In brief, individuals in the infection cohort met the definition of sepsis if they had either ICD codes for septic shock or severe sepsis (ICD-9-CM, 995.92 and 785.52; ICD-10-CM, R65.20 and R65.21) because these are highly specific (99.3%) (*Rhee et al., 2017*), or they met any Sepsis-3 criterion for serious organ dysfunction (*Liu et al., 2023*; *Rhee et al., 2017*). Criteria for organ system dysfunction included: (1) *circulatory*: (a) use of a vasopressor, which we extracted as use of levophed (norepinephrine), or (b) use of the vasopressors (i.e., dobutamine or dopamine) unrelated to stress echocardiography CPT codes 78452, 93015, 93018, 93016, 93017, and 93351 within days –1, 0, and (1) and with ≥2 mentions of any of the keywords (i.e., 'infection,' 'sepsis,' or 'septic'); (2) *respiratory*: ICD or CPT codes for ventilation and admission to an ICU; (3) *renal*: a doubling or greater increase of baseline creatinine (baseline creatinine was defined as the lowest creatinine between 1 year before admission and hospital discharge); (4) *hepatic*: a total bilirubin ≥34.2 μmol/L (2 mg/dL) and at least double from baseline (baseline bilirubin was defined as the lowest total bilirubin occurring between 1 year before admission and hospital discharge); and (5) *hematologic*: a platelet count <100,000/μL and ≥50% decline from a baseline that must have been ≥100,000 (the baseline value was the highest platelet count occurring between 1 year before admission and hospital discharge) (*Liu et al., 2023*; *Feng et al., 2019a*; *Feng et al., 2019b*). Short-term mortality was defined as patients who (1) had death recorded in the EHR within the index hospital stay or (2) were discharged to hospice (*Alrawashdeh et al., 2022*).

## Covariates

We extracted demographic characteristics from the EHRs, including sex and age at the time of the index hospital admission, as well the types of infection at the time of hospitalization (*Supplementary file 1a*). Comorbidities were collected (*MDCalc, 2019*; *National Cancer Data Base - Data Dictionary PUF, 2013*) using relevant diagnostic codes in the year before the index hospital admission (*Supplementary file 1b*) grouped into the 17 Charlson/Deyo comorbidity categories (*Deyo et al., 1992*; *Charlson et al., 1987*; *Quan et al., 2005*). We also identified patients with pre-existing severe renal disease (i.e., Stage 4/5 chronic kidney disease/ESRD as evidenced by one or more of the following ICD diagnosis and procedure codes: N18.4, N18.5, N18.6, N18.9, 585.4, 585.5, 585.6, 585.9, 586, Z99.2, Z49.0, Z49.31, 39.95, V45.11, V56.0, *Supplementary file 1c*). PCs for ancestry were calculated using common variants (minor allele frequency [MAF]>1%) with a high variant call rate (>98%), excluding variants in linkage and regions known to affect PCs (i.e., the HLA region on chromosome 6, inversion on chromosome 8 [8135000–12000000], and inversion on chr 17 [40900000–45000000], GRCh37 build). We calculated 10 PCs for ancestry using SNPRelate version 1.16.0 (*Zheng et al., 2012*).

## Genotyping for *APOL1*

Genotyping was performed using the Illumina Infinium Expanded Multi-Ethnic Genotyping Array (MEGA[EX]). We excluded DNA samples: (1) with a call rate <95%; (2) with inconsistently assigned sex; or (3) that were unexpected duplicates. We performed whole genome imputation using the Michigan Imputation Server (*Das et al., 2016*) with the Haplotype Reference Consortium (*McCarthy et al., 2016*), version r1.1 (*McCarthy et al., 2016*; *Do et al., 2013*), as reference; we then filtered variants with (1) low imputation quality ($r^2$ <0.3), (2) MAF <0.5%, and (3) MAF absolute difference >0.3 when compared to the HRC reference panel.

Variants within *APOL1* were extracted from imputed genotype data. We used rs73885319 to define G1 and rs12106505 as a proxy for the G2 allele (rs71785313) (*Bajaj et al., 2020*). Individuals who were *APOL1* variant allele homozygotes or compound heterozygotes—defined as carriers of two copies of rs73885319 (G1/G1), two copies of rs12106505 (G2/G2), or one copy of each (G1/G2)—were considered to be high risk. Carriers of 1 or 0 *APOL1* risk alleles were considered low risk (i.e., a recessive model) (*Genovese et al., 2010*; *Bajaj et al., 2020*).

## Statistical analysis

Primary and secondary outcomes in patients with high-risk and low-risk *APOL1* genotypes were compared using logistic regression with adjustment for age at hospital admission, sex, and three PCs for ancestry. We performed further analyses (1) with additional adjustment for pre-existing severe renal

disease (*Supplementary file 1c*), (2) excluding patients with pre-existing severe renal disease (n=458) from the infection cohort, and (3) including only patients with pre-existing severe renal disease.

In the replication analysis, we examined selected sepsis-related diagnoses previously reported to be associated with the *APOL1* high-risk genotype in a restricted PheWAS study (*Wu et al., 2021*). We used a similar approach and performed a restricted PheWAS for the selected sepsis-related phenotypes (i.e., infection of internal prosthetic device, phecode 81; septicemia, phecode 38; sepsis, phecode 994.2, systemic inflammatory response syndrome [SIRS], phecode 994.1; and septic shock, phecode 994.21) in all EHR-reported Black patients in BioVU with MEGA$^{EX}$ genotypes (n=14,713). We identified phenotypes using phecodes, a phenotyping system based on ICD-9-CM and ICD-10-CM diagnosis codes (*Wei et al., 2017*; *Wu et al., 2019*). A phecode amalgamates related ICD codes mapping to a distinct disease or trait (*Wei et al., 2017*; *Wu et al., 2019*). A case was defined as an individual with two or more occurrences of the phecode of interest in the EHR. Controls were individuals without that code. Individuals with one mention of the code or with related codes were excluded from the analysis to limit misclassification. We conducted logistic regressions with adjustment for age, sex, and three PCs and, as an expansion of the original approach (*Wu et al., 2021*), repeated the analysis after excluding patients whose EHR contained one or more ICD codes indicating pre-existing severe renal disease (*Supplementary file 1c*).

Chi-square tests were used to compare categorical characteristics and comorbidities between high- and low-risk *APOL1* genotype groups. t-Tests were used to compare continuous characteristics. Logistic regression results are presented as odds ratios (ORs) and 95% confidence intervals; categorical variables are shown as number and percent; continuous variables are shown as median and interquartile range. p-Values <0.05 were considered statistically significant, except the restricted PheWAS for which p-values <0.01 (0.05 divided by 5 phecodes [namely, infection of internal prosthetic device, septicemia, sepsis, SIRS, and septic shock]) were considered significant. All analyses were conducted using R version 4.1.0.

## Results

The primary cohort included 2242 Black patients hospitalized with an infection; 361 (16.1%) patients carried a high-risk *APOL1* genotype and 1881 (83.9%) carried low-risk genotypes (*Table 1*). The baseline characteristics of patients with the high- and low-risk genotypes did not differ significantly in age, sex, and most general medical comorbidities and infection types. However, renal-related comorbidities were significantly more frequent in the high-risk genotype group (p=1.60 × 10$^{-10}$) (*Table 1*).

### Associations between high-risk *APOL1* genotype and sepsis

Within the primary cohort of patients hospitalized with infections, 565 patients developed sepsis, including 105 (29.1%) with *APOL1* high-risk genotypes and 460 (24.5%) with low-risk genotypes. The risk of sepsis was significantly increased among patients with the high-risk *APOL1* genotypes (OR = 1.29 [95% CI, 1.00–1.67; p=0.047]) (*Figure 1*). However, the association between sepsis and *APOL1* high-risk genotypes was not significant after adjustment for pre-existing severe renal disease (OR = 1.14 [95% CI: 0.88–1.48; p=0.33]), nor after exclusion of those patients (n=458) with severe renal disease (OR = 0.99 [95% CI, 0.70–1.39; p=0.95]) (*Figure 2*). We also found no association between sepsis and *APOL1* high-risk genotypes in analysis of patients with severe renal disease alone (OR = 1.29 [95% CI, 0.84–1.98, p=0.25]).

### Associations between the high-risk *APOL1* genotypes and components of sepsis or short-term mortality

For secondary outcomes, among the 565 patients with sepsis, 163 (28.8%) had septic shock, 91 (16.1%) had cardiovascular dysfunction, 136 (24.1%) had respiratory dysfunction, 303 (53.6%) had renal dysfunction, 83 (14.7%) had hepatic dysfunction, 102 (18.1%) had hematologic dysfunction, and 84 (14.9%) died or were discharged to hospice. *APOL1* high-risk genotypes were significantly associated with renal dysfunction component of the Sepsis-3 criteria (OR = 1.64 [95% CI, 1.21–2.22; p=0.001]), but they were not significantly associated with septic shock (OR = 1.30 [95% CI, 0.86–1.95; p=0.21]) nor dysfunction of other organ systems (respiratory: OR = 0.57 [95% CI, 0.33–1.01; p=0.06]; hematologic: OR = 0.86 [95% CI, 0.49–1.51; p=0.60]; circulatory: OR = 0.89 [95% CI, 0.50–1.60;

p=0.70]; hepatic: OR = 1.02 [95% CI, 0.56–1.88; p=0.94]), or short-term mortality (OR = 0.71 [95% CI, 0.31–1.39; p=0.31]) (*Figure 1*, *Supplementary file 1d*).

The association between high-risk *APOL1* genotypes and the renal dysfunction component of the Sepsis-3 criteria was markedly attenuated by adjusting for pre-existing severe renal disease present in the year before the index hospital admission (*Figure 2*, OR = 1.36 [95% CI, 1.00–1.86; p=0.05]) and

**Table 1.** Clinical characteristics of patients admitted to the hospital with infections.

| | | Total (n=2242) | APOL1 low-risk group (n=1881) | APOL1 high-risk group (n=361) | p-Value |
|---|---|---|---|---|---|
| Gender | Female, n (%) | 1306 (58.3) | 1096 (58.3) | 210 (58.2) | |
| | Male, n (%) | 936 (41.7) | 785 (41.7) | 151 (41.8) | 1.00 |
| Age (at admission), median (Q1-Q3) | | 50 (33–62) | 50 (33–63) | 50 (34–60) | 0.67 |
| Comorbidities* | | | | | |
| Congestive heart failure, n (%) | | 312 (13.9) | 260 (13.8) | 52 (14.4) | 0.83 |
| Chronic pulmonary disease, n (%) | | 378 (16.9) | 320 (17) | 58 (16.1) | 0.72 |
| Cerebrovascular disease, n (%) | | 221 (9.9) | 197 (10.5) | 24 (6.6) | 0.03 |
| Dementia, n (%) | | 39 (1.7) | 33 (1.8) | 6 (1.7) | 1.00 |
| Diabetes with chronic complication, n (%) | | 244 (10.9) | 200 (10.6) | 44 (12.2) | 0.44 |
| Diabetes without chronic complication, n (%) | | 170 (7.6) | 136 (7.2) | 34 (9.4) | 0.18 |
| Hemiplegia or paraplegia, n (%) | | 48 (2.1) | 41 (2.2) | 7 (1.9) | 0.93 |
| AIDS/HIV, n (%) | | 44 (2) | 35 (1.9) | 9 (2.5) | 0.56 |
| Malignancy, including lymphoma and leukemia, except malignant neoplasm of skin, n (%) | | 373 (16.6) | 318 (16.9) | 55 (15.2) | 0.48 |
| Myocardial infarction, n (%) | | 222 (9.9) | 189 (10) | 33 (9.1) | 0.67 |
| Mild liver disease, n (%) | | 38 (1.7) | 33 (1.8) | 5 (1.4) | 0.78 |
| Moderate or severe liver disease, n (%) | | 49 (2.2) | 44 (2.3) | 5 (1.4) | 0.35 |
| Peptic ulcer disease, n (%) | | 23 (1) | 19 (1) | 4 (1.1) | 1.00 |
| Peripheral vascular disease, n (%) | | 142 (6.3) | 123 (6.5) | 19 (5.3) | 0.43 |
| Renal disease, n (%) | | 545 (24.3) | 409 (21.7) | 136 (37.7) | 1.60E-10 |
| Rheumatic disease, n (%) | | 73 (3.3) | 58 (3.1) | 15 (4.2) | 0.37 |
| Metastatic solid tumor, n (%) | | 198 (8.8) | 166 (8.8) | 32 (8.9) | 1.00 |
| Infection type† | | | | | |
| Circulatory, n (%) | | 35 (1.6) | 29 (1.5) | 6 (1.7) | 0.86 |
| Digestive, n (%) | | 247 (11) | 217 (11.5) | 30 (8.3) | 0.07 |
| Genitourinary, n (%) | | 769 (34.3) | 648 (34.5) | 121 (33.6) | 0.75 |
| Intestinal, n (%) | | 60 (2.7) | 46 (2.4) | 14 (3.9) | 0.12 |
| Musculoskeletal, n (%) | | 109 (4.9) | 92 (4.9) | 17 (4.7) | 0.89 |
| Neurologic, n (%) | | 43 (1.9) | 35 (1.9) | 8 (2.2) | 0.65 |
| Other bacterial, n (%) | | 879 (39.3) | 723 (38.5) | 156 (43.3) | 0.08 |
| Respiratory, n (%) | | 711 (31.8) | 589 (31.3) | 122 (33.9) | 0.34 |
| Skin, n (%) | | 325 (14.5) | 286 (15.2) | 39 (10.8) | 0.03 |
| Other, n (%) | | 322 (14.4) | 268 (14.3) | 54 (15) | 0.72 |

*Comorbidities present in the preceding year as defined by criteria for the Charlson/Deyo comorbidity index.
†Total n=2239; APOL1 low-risk group n=1879; APOL1 high-risk group n=360.

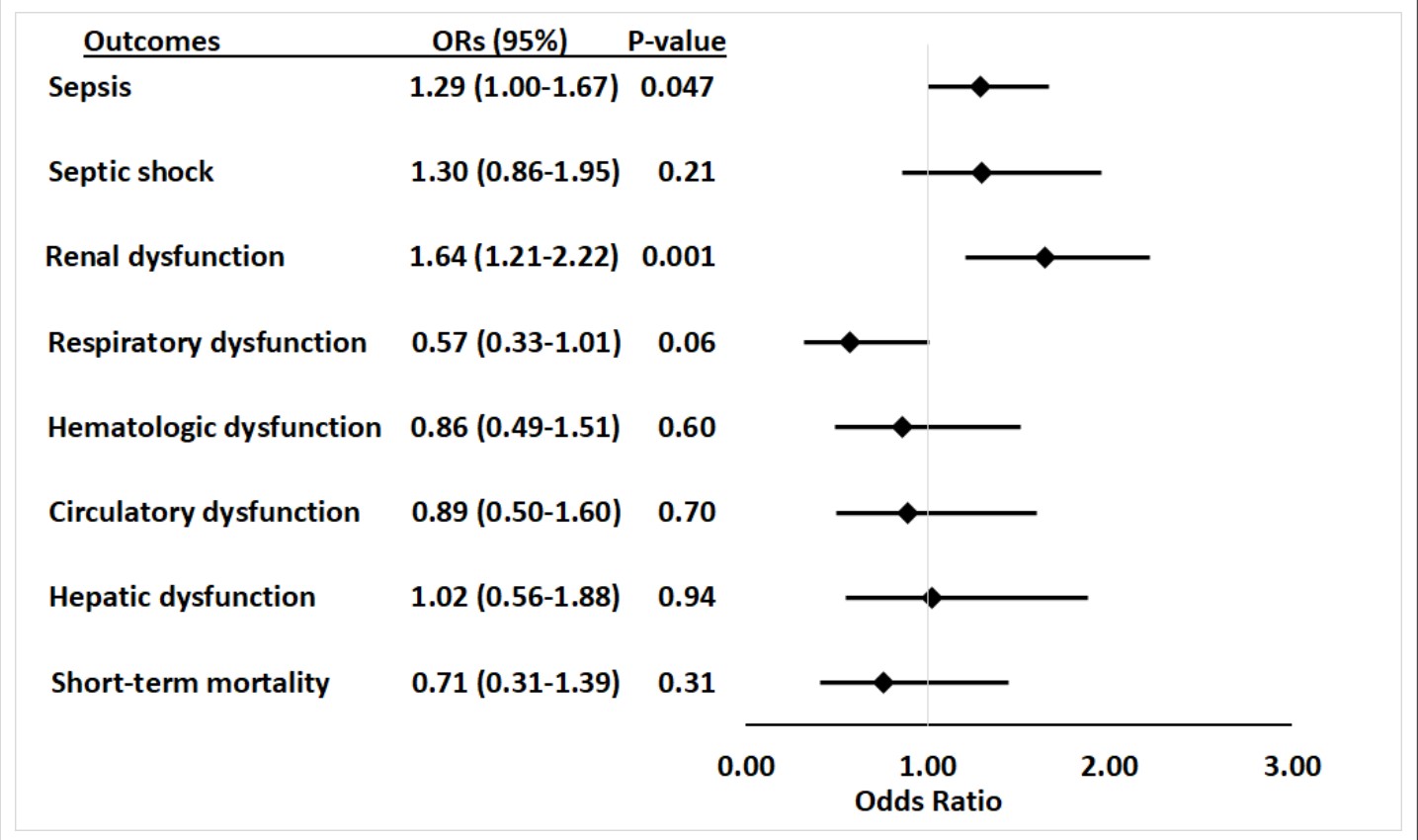

**Figure 1.** Association between *APOL1* high-risk genotypes and the risk of sepsis, and septic shock, and individual sepsis organ dysfunction criteria. Analyses were adjusted for age, sex and three principal components (PCs). Sepsis was defined as meeting the Sepsis-3 criteria for sepsis (***Donnelly et al., 2017***; ***Rhee et al., 2017***), septic shock was defined using International Classification of Disease, ninth revision, Clinical Modification (ICD-9) and -10 codes for septic shock/severe sepsis (ICD-9-CM, 995.92 and 785.52; ICD-10-CM, R65.20 and R65.21); organ dysfunction data represent the individual organ dysfunction criteria in the Sepsis-3 definition; short-term mortality was defined as in-hospital death or discharge to hospice.

The online version of this article includes the following source data for figure 1:

**Source data 1.** Association between APOL1 high-risk genotypes and the risk of sepsis, and septic shock, and individual sepsis organ dysfunction criteria: expanded summary data.

was nullified after the exclusion of patients with pre-existing severe renal disease (***Figure 2***, OR = 1.16 [95% CI, 0.74–1.81; p=0.52]). We also found no association between *APOL1* high-risk genotypes and the renal dysfunction criterion in analysis of patients with severe renal disease alone (OR = 1.43 [95% CI, 0.90–2.26, p=0.13]).

## Associations between *APOL1* high-risk genotype and sepsis-related phenotypes

Using a parallel methodology, the restricted PheWAS performed in Black participants in BioVU (n=14,713, ***Supplementary file 1e***) replicated the association of *APOL1* high-risk genotypes and all prespecified sepsis-related phenotypes previously identified as associated with *APOL1* high-risk genotypes in an MVP cohort: infection of internal prosthetic device, OR = 1.68 (95% CI, 1.32–2.13; p=2.23 × 10⁻⁵); SIRS, OR = 1.49 (95% CI, 1.10–2.01; p=9.87 × 10⁻³); sepsis, OR = 1.41 (95% CI, 1.18–1.67; p=1.30 × 10⁻⁴); septic shock, OR = 1.51 (95% CI, 1.14–1.99; p=3.89 × 10⁻³); and septi-cemia, OR = 1.30 (95% CI, 1.08–1.56; p=6.01 × 10⁻³) (***Figure 3***, panel A). However, in an expansion of the original approach, the associations between *APOL1* and sepsis-related phenotypes were nullified after we excluded individuals with pre-existing severe renal disease (n=2166) and reran the analyses (n=12,547, ***Figure 3***, panel B).

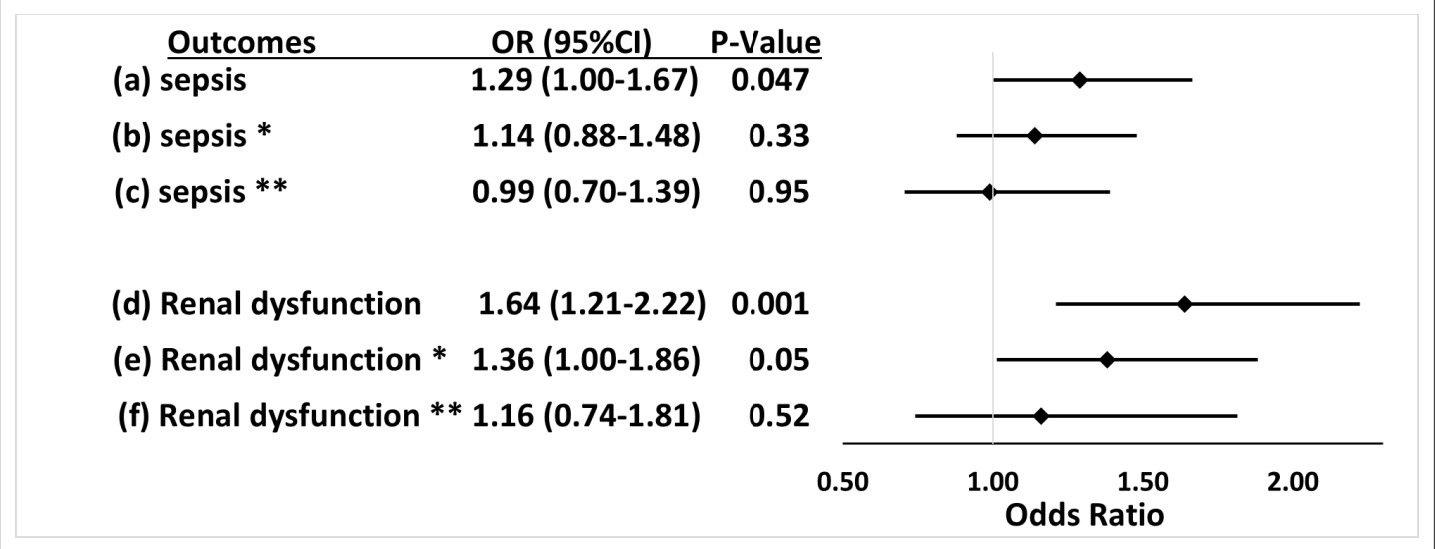

**Figure 2.** Associations between *APOL1* high-risk genotypes and the risk of sepsis and renal dysfunction before and after adjustment for renal disease, and exclusion of patients with pre-existing severe renal disease. We test the associations between *APOL1* high-risk genotypes and the risk of sepsis (**a, b, c**) or the risk of sepsis-related renal dysfunction (**d, e, f**). Sepsis was defined as meeting the Sepsis-3 criteria for sepsis (***Donnelly et al., 2017***; ***Rhee et al., 2017***), sepsis-related renal dysfunction represents the individual organ dysfunction criteria in the Sepsis-S3 definition. Analyses (**a**) and (**d**) were adjusted for age, sex, and three PCs. *In analyses (**b**) and (**e**) we adjusted for age, sex, three principal components, and pre-existing severe renal disease. **In analyses (**c**) and (**f**), we excluded patients with pre-existing severe renal disease and adjusted for age, sex, and three principal components.

The online version of this article includes the following source data for figure 2:

**Source data 1.** Associations between APOL1 high-risk genotypes and the risk of sepsis and renal dysfunction before and after adjustment for renal disease, and exclusion of patients with pre-existing severe renal disease: expanded summary data.

## Discussion

This retrospective cohort study found that *APOL1* high-risk genotypes were significantly associated with an increased risk of sepsis in patients hospitalized with infections; however, this association was explained predominantly by the presence of pre-existing severe renal disease. Renal dysfunction was the only sepsis-associated organ dysfunction significantly associated with *APOL1* high-risk genotypes, and this risk was attenuated by adjustment for pre-existing severe renal comorbidity and nullified by the exclusion of patients with pre-existing severe renal disease. Moreover, there appeared to be no increased risk of sepsis for carriage of high-risk genotypes among patients with pre-existing severe renal disease.

Our findings of an overall increased sepsis risk are consistent with a recent cohort study performed in the MVP which found that high-risk *APOL1* genotypes were associated with ~40% increased risk of sepsis compared to low-risk genotype patients, remaining significant after adjustment for age, sex, and estimated glomerular filtration rate (eGFR) (***Wu et al., 2021***) Additional mechanistic analysis in this study observed that *APOL1* is highly expressed in the endothelium and that the high-risk variants were associated with increased inflammation, endothelial leakage, and sepsis severity (***Wu et al., 2021***). These findings raised the possibility that strategies to inhibit APOL1 in patients who carry the high-risk variants may have therapeutic potential to prevent occurrence or ameliorate symptoms of sepsis.

This study extends and refines the findings of the MVP study, showing an association, if modest, between high-risk *APOL1* genotypes and the occurrence of sepsis among patients admitted to the hospital with infections. However, the association between *APOL1* high-risk genotypes and sepsis is driven largely by the presence of pre-existing severe renal disease—a potent risk factor for infection, sepsis, and infection-related mortality (***Sarnak and Jaber, 2000***; ***Wang et al., 2011***). When we adjusted for pre-existing renal comorbidity, the association between *APOL1* high-risk genotypes and sepsis was attenuated, and when we removed patients with pre-existing severe renal disease from the analysis, the significant association was nullified. Additionally, in the restricted PheWAS of all Black participants in BioVU (a design parallel to that of the MVP study), we found that *APOL1* high-risk

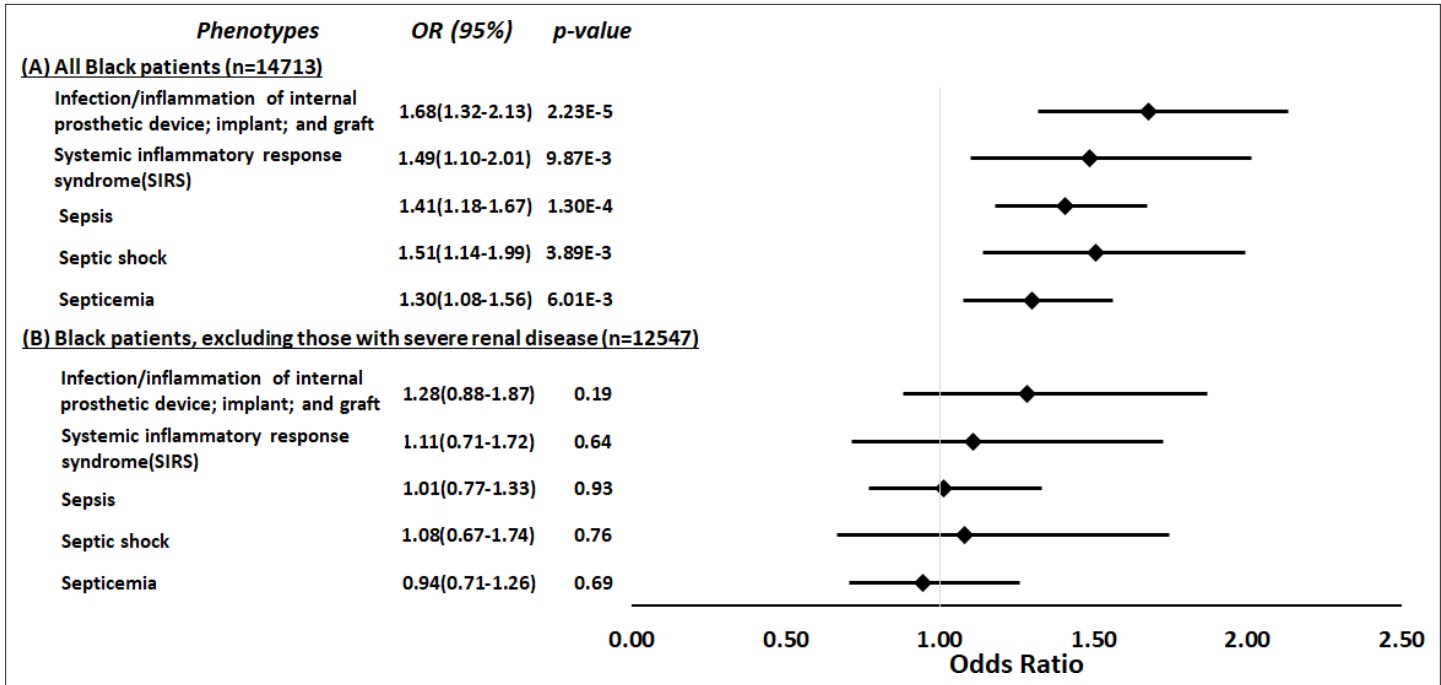

**Figure 3.** Restricted phenome-wide association analysis (PheWAS) associations between *APOL1* high-risk genotype and sepsis-related phenotypes. We tested the associations between *APOL1* high-risk genotype and sepsis-related phenotypes, including infection of internal prosthetic device, systemic inflammatory response syndrome (SIRS), sepsis, septic shock, and septicemia. We conduct the analyses in all electronic medical record (EHR)-reported Black individuals with available MEGA^EX genotypes in BioVU (n=14,713, panel A) and after excluding individuals with severe renal diseases (n=12,547, panel B). Analyses were adjusted for age, sex, and three principal components.

The online version of this article includes the following source data for figure 3:

**Source data 1.** Restricted phenome-wide association analysis (PheWAS) associations between APOL1 high-risk genotype and sepsis-related phenotypes: expanded summary data.

genotypes were significantly associated with all prespecified sepsis-related phenotypes. However, those associations were not significant after excluding patients with severe renal disease.

Three differences in study design may contribute to differences in the findings of the two studies. First, the retrospective cohort approach allowed us to better define the temporal relationship between comorbidities and sepsis and its organ dysfunction criteria. Second, we adjusted for renal disease rather than eGFR, because in the setting of dialysis or renal transplantation, the GFR estimated from a creatinine measurement may not capture the increased risk of sepsis in these patients. Third, we used a validated EHR algorithm rather than phecodes to define sepsis; nevertheless, the replication analysis using phecodes was consistent with the primary analysis—*APOL1* high-risk genotypes were associated with sepsis-related phecodes; however, these associations were driven by those patients with pre-existing severe renal disease, a known consequence of *APOL1* high-risk genotypes.

The lack of association independent of pre-existing severe renal disease suggests that *APOL1* high-risk genotypes are not acutely causal of sepsis beyond their association with renal disease and impaired renal function (which, in turn, increases susceptibility to sepsis). These findings more closely parallel those of another MVP study that excluded patients with severe pre-existing renal dysfunction and examined the effects of *APOL1* genotypes on outcomes among patients hospitalized with COVID-19 infections. In that study, high-risk *APOL1* genotypes were more strongly associated with acute kidney injury (**Hung et al., 2022**) than the need for mechanical ventilation or vasopressors. These results suggest that drug therapies in development to prevent and treat diseases associated with *APOL1* high-risk genotypes might primarily affect the renal vulnerabilities that increase risk of sepsis, rather than immediate prevention of sepsis or acute treatment of sepsis once hospitalized.

The current study offers several strengths. First, we identified patients progressing from infection to sepsis using a validated EHR algorithm. This approach also allowed us to evaluate each organ dysfunction and its contribution to sepsis separately. Second, by leveraging the rich longitudinal EHRs, we

were able to identify pre-existing severe renal disease and assess its effect on the risk of developing sepsis. Third, we performed analyses that included, excluded, and exclusively focused on patients with pre-existing chronic severe renal disease, allowing us to more completely define the contribution of *APOL1* high-risk genotypes to the development of sepsis.

We also acknowledge the study's limitations, primarily related to a retrospective cohort study using EHR information. First, ascertainment of comorbidities was based on ICD codes, and these may not completely reflect comorbidities. Second, there are many factors that affect health (e.g., alcohol use, diet, and lifestyle) that are not captured well in the EHR but could impact susceptibility to sepsis; however, there is no reason to expect that such factors are differentially distributed according to genotype. Third, we defined death as short-term mortality, including in-hospital death and discharge to hospice (*Alrawashdeh et al., 2022*); this definition may underestimate the true mortality rate due to sepsis. Fourth, we did not include patients with concurrent COVID-19 infections because their clinical manifestations and genetic predispositions might differ from that of patients who develop sepsis after presumed bacterial infection. Fifth, as we did not find an association after adjustment for pre-existing severe renal disease, we did not perform any additional functional analysis. Last, a larger cohort of Black patients with infection and sepsis could potentially provide more power such that the confidence intervals around the point estimates were smaller, more definitively excluding the possibility of clinically important differences in outcomes; a larger cohort could also provide the opportunity for additional granularity in analysis (e.g., stratified by whether patients are on dialysis).

In conclusion, in this cohort of Black participants hospitalized with infection, *APOL1* high-risk genotypes were associated with an increased risk of sepsis; however, this increased risk was attributable predominantly to pre-existing severe renal disease. Further, renal dysfunction was the only sepsis-associated organ dysfunction associated with *APOL1* high-risk genotypes.

## Acknowledgements

The first and corresponding authors had full access to all the data in the study and take responsibility for the integrity of the data and the accuracy of the data analysis.

## Additional information

### Funding

| Funder | Grant reference number | Author |
|---|---|---|
| National Institute of General Medical Sciences | R01GM120523 | Qiping Feng |
| National Heart, Lung, and Blood Institute | R01HL163854 | Qiping Feng |
| National Institute of General Medical Sciences | R35GM131770 | C Michael Stein |
| National Heart, Lung, and Blood Institute | HL133786 | Wei-Qi Wei |

The funders had no role in study design, data collection and interpretation, or the decision to submit the work for publication.

### Author contributions

Lan Jiang, Conceptualization, Data curation, Formal analysis, Investigation, Visualization, Methodology, Writing – original draft, Writing – review and editing; Ge Liu, Data curation, Formal analysis, Writing – review and editing; Annette Oeser, Data curation, Project administration, Writing – review and editing; Andrea Ihegword, Laura L Daniel, Adriana M Hung, Cecilia P Chung, Data curation, Methodology, Writing – review and editing; Alyson L Dickson, Data curation, Formal analysis, Investigation, Visualization, Methodology, Writing – original draft, Writing – review and editing; Nancy J Cox, Data curation, Software, Visualization, Methodology, Writing – review and editing; Wei-Qi Wei, Data curation, Software, Funding acquisition, Investigation, Methodology, Writing – original draft, Writing

– review and editing; C Michael Stein, Conceptualization, Data curation, Supervision, Funding acquisition, Methodology, Writing – original draft, Writing – review and editing; Qiping Feng, Conceptualization, Data curation, Supervision, Funding acquisition, Investigation, Methodology, Writing – original draft, Writing – review and editing

## Author ORCIDs
Alyson L Dickson ⓘ https://orcid.org/0000-0003-3404-3802
Laura L Daniel ⓘ https://orcid.org/0000-0003-3143-5915
Qiping Feng ⓘ http://orcid.org/0000-0002-6213-793X

## Ethics
This study was reviewed by the VUMC Institutional Review Board; given the study's retrospective design and use of deidentified data only, informed consent was waived.

Joint Public Review: https://doi.org/10.7554/eLife.88538.3.sa1
Author Response https://doi.org/10.7554/eLife.88538.3.sa2

## Additional files

### Supplementary files
• Supplementary file 1. Supplementary tables. (a) International Classification of Disease, ninth revision, Clinical Modification (ICD-9-CM) and ICD-10-CM codes for infection categories. (b) ICD-9-CM and ICD-10-CM codes for comorbidities. (c) List of codes to identify severe renal diseases. (d) Associations between *APOL1* high-risk genotypes and sepsis-related outcomes. (e) Demographic summary of patients in the phenome-wide association analysis (PheWAS) analysis.

• Supplementary file 2. STROBE checklist.

• MDAR checklist

### Data availability
Statistical code is available at https://github.com/FengLabVUMC/APOL1_Sepsis (copy archived at *FengLabVUMC, 2023*). Data set: VUMC SD data are de-identified using Safe-harbor methods. BioVU genomic data are linked to de-identified records and are further protected by BioVU data use agreements ensuring that researchers will not attempt re-identification. BioVU participant consenting procedures limit access to individual level data. Limited primary cohort data are available by request to Dr. Feng (e-mail, Qiping.feng@vumc.org), pending BioVU approval and a data use agreement.

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
