## [Editor Report · eLife assessment]

In this **valuable** study, patients homozygous for both minor frequency alleles of the APOL1 gene are shown to be at significant risk for progression into sepsis after infection. The study has enrolled a significant number of subjects and provides **solid** results. The study addresses to infectious diseases and critical care experts and one major weakness is the lack of inclusion of non-Black patients.

---

## [Referee Report · Joint Public Review]

The study has many limitations which need to be addressed and there is a lack of functional explanation of carriage. These limitations are: (a) the lack of inclusion of non-Black patients; and (b) the lack of appropriate explanation if results are false-positive since APOL1 provides risk for chronic renal disease (CRD) and patients with CRD are predisposed to sepsis. Sepsis occurred in 565 Black subjects, of whom 105 (29% ) had APOL1 high-risk genotype and 460 had-low risk genotype. Importantly, the risk for sepsis associated with APOL1 HR variants was no longer significant after adjusting for subjects pre-existing severe renal disease or after excluding these subjects. Thus, the susceptibility pathway seems to be: APOL1 variants > CKD > sepsis diathesis.

---

## [Author Response]

The following is the authors’ response to the original reviews.

Thank you for the helpful comments regarding our manuscript, "Association between APOL1 risk variants and the occurrence of sepsis among patients hospitalized with infections.” We have revised the title of the manuscript in response to reviewer comments. Additionally, we have updated the manuscript with analyses among patients with pre-existing renal disease alone as well as other items suggested by the reviewers. The Tables have been renumbered to accommodate these revisions.

**Public review:**
The study has main limitations which need to be addressed and there is lack of functional explanation of carriage. These limitations are: (a) the lack of inclusion of non-Black patients; and (b) the lack of appropriate explanation if results are false-positive since APOL1 provides risk for chronic renal disease (CRD) and patients with CRD are predisposed to sepsis. Sepsis occurred in 565 Black subjects, of whom 105 (29% ) had APOL1 high-risk genotype and 460 had-low risk genotype. Importantly, the risk for sepsis associated with APOL1 HR variants was no longer significant after adjusting for subjects pre-existing severe renal disease or after excluding these subjects. Thus, the susceptibility pathway seems to be: APOL1 variants > CKD > sepsis diathesis.Suggestions to the authors:• The authors need to provide analysis of patients of non-Black origin.

We apologize for not fully clarifying that the APOL1 high-risk genotypes are virtually exclusive to populations of recent African ancestries,1–4 the majority of whom are identified as having Black race in our dataset.5 To illustrate the rarity of APOL1 high-risk genotypes in other reported races, we examined the frequency of these genotypes in White patients who had been hospitalized with infections at VUMC (comparable to the cohort of Black patients used in the study). Compared to the 361 out of 2242 (16.1%) Black patients hospitalized with infections carrying APOL1 high-risk genotypes, there were only 8 carriers of APOL1 high-risk genotypes out of 12,990 White patients (0.06%); of these 8, 2 patients developed sepsis during hospitalization. Due to a low number of carriers (n=8) and limited number of events (n=2), we could not proceed with further analysis. Patients reported as other races (e.g., Asian and American Indian) are less frequent than White or Black patients in the VUMC de-identified EHR; as such, we would anticipate similarly small, if any, numbers of high-risk genotypes among these groups, with insufficient power for meaningful analysis. Comparisons between racial groups that did not have carriage of the APOL1 high-risk genotypes would increase the possibility of confounding by factors associated with racial identity (e.g., social determinants of health), rather than genotype; as such, detected differences would likely reflect those factors, rather than the impact of APOL1.

We have now added clarifying language in the Methods section.

• The Table of demographics needs to include the type of infections and the underlying pathogen.

Microbiological evidence of specific infection types is not available for the majority of records for patients hospitalized with infections (as well as sepsis); indeed, for many patients with common infections (e.g., pneumonia) the pathogen is often not identified.6 While we do not have details regarding the underlying pathogens, we were able to determine infection categories at admission. We now include details regarding the categories of infection based on ICD codes in Supplementary Table 1, and the updated Table 1 now includes that information for the APOL1 high-risk and low-risk groups. Given that individuals could have more than one type of infection, we also tested the number of types of infection and found no significant difference between the high-risk and low-risk genotypes (p=0.77).

• The authors need to provide convincing analysis if results are false-positive since APOL1 provides risk for chronic renal disease (CRD) and patients with CRD are predisposed to sepsis. For this purpose, they have to provide evidence if the sepsis causes (both type of infection and implicated pathogens) in patients with CRD who are carriers of APOL1 variants are different than in patients with CRD who are not carriers of APOL1 variants.

Indeed, we believe the presented findings suggest that the apparent association between APOL1 high-risk genotypes and sepsis is driven by associated pre-existing severe renal disease rather than APOL1 itself; we appreciate the suggestion to conduct additional analyses to assess whether APOL1 high-risk genotypes impact the occurrence of sepsis among those patients with pre-existing severe renal disease. We note that this analysis could also be biased towards detecting a spurious association between APOL1 high-risk genotypes and sepsis if, within the subgroup with pre-existing severe renal disease, patients with high-risk genotypes also have more severe pre-existing renal disease.

Among the patients with pre-existing severe renal disease (n=458), 121 (26.4%) were carriers of the APOL1 high-risk genotypes. First, we assessed the severity of renal disease among these patients, detecting an association between APOL1 high-risk genotypes and greater severity (i.e., CKD stage 5/ESRD) when adjusted for age, sex, and 3 PCs: OR=2.29 (95% CI, 1.42-3.67, p=6.25x10-4). Then, we compared the primary outcome of sepsis in patients with APOL1 high-risk and low-risk genotypes for this subgroup. Despite the potential bias toward detecting an association between sepsis and the high-risk genotype based on the severity of pre-existing renal disease, there was no significant association between the high-risk genotypes and sepsis (OR=1.29, [95% CI, 0.84-1.98, p=0.25]). Finally, we assessed infection categories (as described in the above response) in this subgroup. We found no significant differences between the high-risk and low-risk genotypes in the frequency of any infection category.

These results suggest that the APOL1 high-risk genotypes are not associated with an increased risk of sepsis among patients who have pre-existing severe renal disease. Taken with our other findings, the high-risk genotypes appear to have little or no association with sepsis beyond their association with renal disease. As such, drugs targeting those genotypes would likely have little effect in the acute setting of hospitalization with infection; rather, their primary contribution to the prevention of sepsis would need to target the prevention of underlying renal disease. We have revised our Methods, Results, and Discussion to include these findings.

• Why concentrations of APOL1 were not measured in the plasma of patients?

Although APOL1 high risk genetic variants have been repeatedly associated with renal-related clinical phenotypes, and many candidate mechanisms have been proposed,4 there has been contradictory evidence regarding whether the genetic variants could be linked to altered plasma APOL1 levels or whether APOL1 levels are related to elevated risk of renal disease. This is not surprising since it is the altered biological function of the APOL1 structural variant that is important, rather than the concentration of APOL1 protein. While some studies have detected an association between APOL1 high-risk genotypes and plasma levels among patients with renal disfunction and sepsis,7 other population studies have suggested no association between APOL1 plasma levels and renal function.8 Plasma APOL1 levels are seldom measured in clinical practice and thus were not available in this retrospective cohort. However, given the inconsistency of findings and the underlying biology of APOL1, we believe measurements of levels (rather than function) is unlikely to be illuminating.

• Why analysis towards risk for death is not done?

In the current study, we focused on the risk of in-hospital death. We did not include the risk of out-of-hospital death due to potential data fragmentation. Specifically, we only have access to the patient’s EHRs at VUMC, and death after hospital discharge is not always be included in a patient’s EHR unless relatives contact the hospital. As such, we focused on in-hospital death, which we validated previously with manual chart review.9 Paralleling the design from a previous publication assessing sepsis outcomes, we included discharge to hospice as part of our in-hospital death algorithm,10 as patients with a terminal illnesses are often discharged to hospice. However, to clarify this outcome component, we now refer to in-hospital deaths and discharge to hospice collectively as “short-term mortality.” In this study, of the 84 total patients with the “short-term mortality” outcome, 47 patients were in-hospital deaths and 37 patients were discharged to hospice. Parallel to the short-term mortality, we found no association with in-hospital death alone.Ln 190: discharge to hospice. I am not sure this can be translated in in-hospital mortality.As noted in the above response, we have rephrased this outcome component as “short-term mortality,” following the design of a previous publication assessing sepsis outcomes.10

• The authors need to explain why functional information is not provided.

Functional studies were not performed for several reasons. Animal models are problematic because mice do not have an ortholog to the human APOL1 gene, and the various models developed all have limitations, particularly when second and third perturbations (sepsis and renal impairment) would need to be introduced.11 Also, since we did not observe an association between the genotypes and sepsis independent of pre-existing severe renal disease, we did not pursue additional functional studies. We do describe existing functional analysis in the introduction and briefly in our discussion; we now note this limitation.

• n 162-172: too many assumptions have been used for the trial; thus, progression to sepsis is difficult to define. According to Sepsis-3 sepsis is no more a continuum from infection to sepsis and septic shock. Some patients presented with sepsis (-1, 0, 1 days considered by the authors) and when electronic health records are used, we are not able to detect the exact timepoint of SOFA score turning to a 2-point increase. This is a major limitation of the methodology presented.Same applies for all comorbidities and data extracted from electronic health records.

Thank you for highlighting this issue. We acknowledge that our choice of wording was unclear. The choice of ICD infection codes during the initial hospitalization window (i.e., -1, 0, 1 days) was aimed to generate a clean cohort of patients hospitalized with infections (i.e., not secondary infections or development of sepsis after an in-hospital procedure), rather than to establish a timeline of progression from infection to sepsis. As you correctly note, our algorithm would capture patients presenting with infection and concurrent sepsis at admission rather than progression to sepsis, and the exact timepoint of the SOFA score meeting the 2-point criterion is difficult to capture through the EHR. Accordingly, we conducted no time-dependent analysis in the current study. To more accurately convey the methodology of the current study (i.e., testing the association between APOL1 high-risk genotypes—which the patients were born with—and the risk of sepsis for patients hospitalized with infections), we revised the manuscript thoroughly, replacing “progression to sepsis” with “occurrence of sepsis” in the title, abstract as well as on pages 7, 8, and 19.We also acknowledge the limitations of using EHR in the Discussion.

• P value significance thresholds were set at 0.05, except for the PWAS where the threshold was set at 0.05/5 (p13). It would be helpful to list at this point what the 5 outcomes were that led to this adjusted threshold.

We have revised the manuscript accordingly.

"Risk of sepsis was significantly increased among patients with high-risk genotypes (OR 1.29, 1.0 to 1.67, P1.29, CI 1.00-1.67, P<0.47)." Some would argue that a confidence interval that includes 1.0 indicates non-significance.

While the lower bound of the confidence interval appears to meet the 1.0 threshold with only 2 decimal places (which would preclude significance), when taken to the 4th decimal place, the value is 1.0037, demonstrating that the 95% CI did not meet or cross under the 1.0 threshold, and thus the odds ratio should be considered significant (as evidenced by the p=0.047). This result is consistent with other studies that have detected an association between the high-risk genotypes and sepsis,7 but you correctly note that readers can discern from the confidence intervals that the finding is not strong.

• The Discussion is too long and should be shortened.

We have revised the Discussion.

References:

1. Limou S, Nelson GW, Kopp JB, Winkler CA. APOL1 Kidney Risk Alleles: Population Genetics and Disease Associations. Adv Chronic Kidney Dis. 2014;21(5):426-433. doi:10.1053/j.ackd.2014.06.005

2. Kopp JB, Nelson GW, Sampath K, et al. APOL1 genetic variants in focal segmental glomerulosclerosis and HIV-associated nephropathy. J Am Soc Nephrol. 2011;22(11):2129-2137. doi:10.1681/ASN.2011040388

3. Zhang J, Fedick A, Wasserman S, et al. Analytical Validation of a Personalized Medicine APOL1 Genotyping Assay for Nondiabetic Chronic Kidney Disease Risk Assessment. The Journal of Molecular Diagnostics. 2016;18(2):260-266. doi:10.1016/j.jmoldx.2015.11.003

4. Daneshpajouhnejad P, Kopp JB, Winkler CA, Rosenberg AZ. The evolving story of apolipoprotein L1 nephropathy: the end of the beginning. Nat Rev Nephrol. 2022;18(5):307-320. doi:10.1038/s41581-022-00538-3

5. Dumitrescu L, Ritchie MD, Brown-Gentry K, et al. Assessing the accuracy of observer-reported ancestry in a biorepository linked to electronic medical records. Genet Med. 2010;12(10):648-650. doi:10.1097/GIM.0b013e3181efe2df

6. Wiese AD, Griffin MR, Stein CM, et al. Validation of discharge diagnosis codes to identify serious infections among middle age and older adults. BMJ Open. 2018;8(6):e020857. doi:10.1136/bmjopen-2017-020857

7. Wu J, Ma Z, Raman A, et al. APOL1 risk variants in individuals of African genetic ancestry drive endothelial cell defects that exacerbate sepsis. Immunity. 2021;54(11):2632-2649.e6. doi:10.1016/j.immuni.2021.10.004

8. Kozlitina J, Zhou H, Brown PN, et al. Plasma Levels of Risk-Variant APOL1 Do Not Associate with Renal Disease in a Population-Based Cohort. J Am Soc Nephrol. 2016;27(10):3204-3219. doi:10.1681/ASN.2015101121

9. Liu G, Jiang L, Kerchberger VE, et al. The relationship between high density lipoprotein cholesterol and sepsis: A clinical and genetic approach. Clin Transl Sci. 2023;16(3):489-501. doi:10.1111/cts.13462

10. Alrawashdeh M, Klompas M, Simpson SQ, et al. Prevalence and Outcomes of Previously Healthy Adults Among Patients Hospitalized With Community-Onset Sepsis. Chest. 2022;162(1):101-110. doi:10.1016/j.chest.2022.01.016

11. Yoshida T, Latt KZ, Heymann J, Kopp JB. Lessons From APOL1 Animal Models. Front Med (Lausanne). 2021;8:762901. doi:10.3389/fmed.2021.762901